# Exploring Data Augmentation and Dimension Reduction Opportunities for Predicting the Bandgap of Inorganic Perovskite through Anion Site Optimization

**Tri-Chan-Hung Nguyen [1], Young-Un Kim [1], Insung Jung [2], O-Bong Yang [1,2] and Mohammad Shaheer Akhtar [1,2,3,*]**

1   Graduate School of Integrated Energy-AI, Jeonbuk National University, Jeonju 54896, Republic of Korea; ntchung3397@jbnu.ac.kr (T.-C.-H.N.)
2   New & Renewable Energy Material Development Center (NewREC), Jeonbuk National University, Jeonju 56332, Republic of Korea
3   Department of JBNU-KIST Industry-Academia Convergence Research, Jeonbuk National University, Jeonju 54896, Republic of Korea
*   Correspondence: shaheerakhtar@jbnu.ac.kr

**Abstract:** Significant focus has been directed towards inorganic perovskite solar cells because of their notable capabilities in converting sunlight to electricity effectively, their efficient light absorption, and their suitability for conventional semiconductor manufacturing methods. The identification of the composition of perovskite materials is an ongoing challenge to achieve high performing solar cells. Conventional methods of trial and error frequently prove insufficient, especially when confronted with a multitude of potential candidates. In response to this challenge, the suggestion is to employ a machine-learning strategy for more precise and efficient prediction of the characteristics of new inorganic perovskite materials. This work utilized a dataset sourced from the Materials Project database, consisting of 1528 $ABX_3$ materials with varying halide elements (X = F, Cl, Br, Se) and information regarding their bandgap characteristics, including whether they are direct or indirect. By leveraging data augmentation and machine learning (ML) techniques along with a collection of established bandgap values and structural attributes, our proposed model can accurately and rapidly predict the bandgap of novel materials, while also identifying the key elements that contribute to this property. This information can be used to guide the discovery of new organic perovskite materials with desirable properties. Six different machine learning algorithms, including Logistic Regression (LR), Multi-layer Perceptron (MLP), Decision Tree (DT), Support Vector Machine (SVM), Extreme Gradient Boosting (XGBoost), and Random Forest (RF), were used to predict the direct bandgap of potential perovskite materials for this study. RF yielded the best experimental outcomes according to the following metrics: F1-score, Recall, and Precision, attaining scores of 86%, 85%, and 86%, respectively. This result demonstrates that ML has great potential in accelerating organic perovskites material discovery.

**Keywords:** perovskite; bandgap; optimization; feature selection; machine learning

## 1. Introduction

Three-dimensional (3D) inorganic perovskite materials have gained significant attention in the realm of renewable energy research due to their distinctive structural and optoelectronic properties. Among them, perovskite solar cells (PSCs) are one of the most advanced applications, with their power conversion efficiencies (PCE) having increased rapidly from 3.5% [1] to 26.1% in a decade [2–4]. In addition to this, PSCs have some distinct advantages such as low-cost raw materials, low processing cost, simple manufacture, and flexible fabrication. These materials, characterized by the $ABX_3$ crystal structure, consist of corner-sharing $BX_6$ octahedra, creating a versatile framework for tuning properties by altering the constituents A, B, and X [5]. This structural flexibility offers opportunities to

tailor their bandgap, a critical parameter that dictates their performance in photovoltaic applications [6–9]. Traditionally, predicting the bandgap nature of materials like 3D inorganic perovskites relied on empirical rules and computational methods rooted in density functional theory (DFT) [10]. DFT has provided valuable insights into electronic structure, such as its accuracy in predicting bandgaps. Nevertheless, DFT requires many computing resources and an understanding of quantum chemistry, and the weakness of this method on complex materials can be traced back to two main errors of the standard density functional: the delocalization error and the static correlation error [11,12].

This difference necessitates innovative approaches to improving bandgap prediction accuracy. Machine learning, a developing field, has demonstrated promise in addressing this challenge [13]. By training models on diverse datasets of material properties, machine learning algorithms can capture complex correlations between structural features and bandgap values, thereby enabling more accurate predictions [14]. Data augmentation, which involves generating synthetic data to augment training sets, further enhances model robustness and generalization. Several recent studies have highlighted the potential of machine learning in predicting bandgaps of 3D inorganic perovskites. For instance, Rosen et al. [15] employed several ML models trained on 14,000 MOF structures (the QMOF database) to accurately predict bandgaps of diverse perovskite compositions. In addition, Kumar et al. [16] utilized a convolution neural network-based gradient-boosting framework to predict the bandgap of photoactive catalysts. Not only providing the prediction of bandgap using machine learning, Liu et al. [17] provided an experiment to explain the hidden relationship between each feature of the bandgap values in a specific dataset. From the success of bandgap prediction, much research and many applications have used machine learning to discover the materials in optoelectronic devices [18–20]. A highlight of the contributions presented by Zhang et al. [21] resides in the utilization of diverse machine learning methodologies to define possible candidates among halide perovskites intended for solar cell utilization. This approach tackles common concerns about the absence of lead content and the stability challenges that have constituted pivotal concerns within the domain of perovskite solar cell technology. X. Cia et al. [22] reported on the optimization of materials' discovery by applying ML algorithms to unveil the connection between critical parameters and photovoltaic performance with high-profile $MASn_xPb_{1-x}I_3$ perovskite materials. Another work by Q. Tao et al. [23] described different ML algorithms to identify different properties of inorganic, hybrid organic-inorganic, and double perovskites to optimize the experiment process during the property's discovery of new materials. Work reported by J. Li et al. [24] employed a dataset comprising 760 perovskites to determine the phonon cutoff frequency and applied six distinct ML algorithms to predict this instrumental variable using features available within the provided database.

The contribution of this study mainly consists of four different aspects:

(1) Data processing has been approached to clean, null, and duplicate values from 1528 materials. This dataset was characterized by an intricate feature space spanning 130 dimensions, encompassing pertinent attributes including the nature of the bandgap.

(2) Through the raw database, the application of the data augmentation methodology was undertaken to enhance the diversity of the dataset. Additionally, the implementation of Pearson Correlation was performed to ascertain the degree of correlation between each individual feature, and, notably, the intrinsic nature of the bandgap, thereby uncovering latent relationships.

(3) Following the completion of the data processing pipeline, the refined dataset underwent training across a spectrum of six distinct machine learning algorithms. The selection process was driven by the pursuit of optimal performance as gauged by precision, recall, and F1-score [25] criteria.

(4) Discovering the predictive mechanisms of ML models opens up unexpected findings that can be used to further research directions. By applying game theory to assign credit for the model's prediction to each feature value based on Shapley Additive

exPlanations (SHAP) [26] algorithms, these values can be used to understand the importance of each feature, explain the result of the machine learning model, and represent essential features that can directly affect the natural bandgap. Finally, the investigation reveals that variations in the range neighbor distance hold paramount importance in determining the character of the bandgap, surpassing the influence of other feature values.

## 2. Materials and Methods

As shown in Figure 1, this research is structured into three main parts: Construction of the dataset, Data Processing, and Modeling.

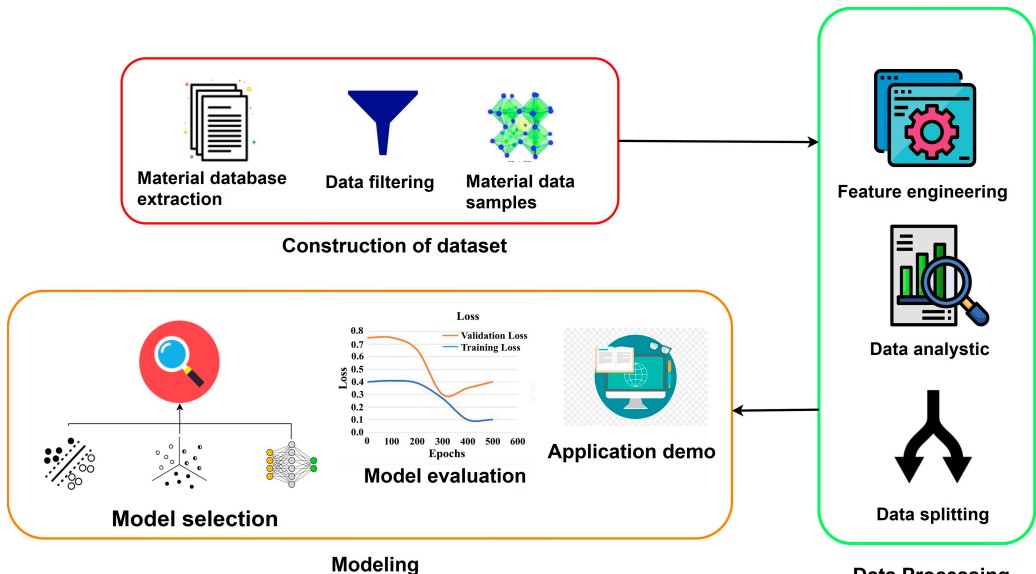

**Figure 1.** The workflow of this work to optimize the perovskite bandgap.

- Construction of the dataset: This process involves extracting data from an open-source database and filtering it to identify relevant features related to the materials from the raw data.
- Data Processing: This phase is crucial for understanding hidden relationships and identifying important features in the ML algorithms based on numeric information.
- Modeling: This step entails selecting a suitable ML algorithm for the dataset and conducting the final experiments based on the collective results from the previous stages.

The methodology adopted for this research is designed to comprehensively investigate and predict the direction of the bandgap in materials established with model explanation, data generating, data engineering, and model experiments.

### 2.1. Data Construction

The dataset employed in this study originates from the Materials Project and was collected by Smarak Rath et al. [27]. in 2022. The dataset encompasses 1528 distinct chemical compounds with the $ABX_3$ formula. Each row within the dataset includes information about the nature of the bandgap and the chemical formula, as well as 130 distinct features associated with parameters such as molecular distance, valence electrons, and bond length between diverse molecules. The compilation of these materials was facilitated using Pymatgen [28], a Python library package provided by the Material Project Database [29]. The structural attributes of the materials obtained through Pymatgen contain details regarding unit cell parameters and atomic placements within the unit cell.

Given the presence of a dataset featuring a multitude of high-dimensional features, the systematic exploration of the dataset becomes an essential process to discover data patterns,

duplicate information, and missing values in the dataset. Based on this, the utilization of data summarization techniques has been applied to encapsulate comprehensive insights concerning the dataset. The sample result of these techniques is documented in Table 1. In addition to the sample shown in Table 1, the entire dataset in this study has been validated to have no missing or duplicate entries during the data preprocessing phase.

**Table 1.** Dataset summarization of 5 features from the total 130 features.

| Name | Data Types | Missing | Uniques | Mean | Standard Deviation |
|---|---|---|---|---|---|
| Nature of bandgap | Int64 | 0 | 2 | 0.270 | 0.444 |
| Max relative bond length | Float64 | 0 | 1368 | 1.095 | 0.041 |
| Min relative bond length | Float64 | 0 | 1369 | 0.806 | 0.069 |
| Frac s valence electrons | Float64 | 0 | 114 | 0.416 | 0.454 |
| Frac p valence electrons | Float64 | 0 | 191 | 0.583 | 0.545 |

### 2.2. Data Engineering

Data engineering is an extremely important process after summarizing and cleaning the data. Within this stage, data augmentation and feature selection are proposed to tackle the challenges posed by unbalanced datasets. Moreover, the removal of redundant features is undertaken to enhance the performance of machine learning models in terms of both speed and accuracy. The below subsection elucidates the benefits of employing these techniques.

#### 2.2.1. Data Augmentation

This research confronts the challenge of an imbalanced dataset, comprising 884 samples labeled as "non-natural bandgap" and only 340 samples labeled as "nature of the bandgap." This skewed distribution poses a significant problem as the underrepresentation of the "nature of the bandgap" class could result in biased model training and hinder accurate predictions for classification tasks. To address this issue, Synthetic Minority Oversampling Technique (SMOTE) [30] emerges as a compelling solution. SMOTE leverages an ingenious approach to tackle class imbalance by generating synthetic instances of the minority class ("nature of the bandgap") based on the existing data distribution. By doing so, SMOTE effectively bridges the gap in class representation, augmenting the dataset and leveling the playing field for the model. The SMOTE works by generating synthetic examples of the minority class by interpolating between existing minority class instances. SMOTE selects a minority class instance, identifies its k-nearest neighbors, and creates synthetic examples by blending the features of the selected instance with those of its neighbors which is visualized in Figure 2. This augmentation process entails creating synthetic samples in feature space regions where the minority class is underrepresented, effectively enriching the dataset, and rectifying the imbalance. The general formula of SMOTE can be explained below:

$$SI = x_i + \text{rand} * (x_{nn} - x_i) \tag{1}$$

where:

$\quad$ SI: the generated synthetic instance.
$\quad$ $x_i$: the minority class instance
$\quad$ $x_{nn}$: a randomly selected nearest neighbor from k neighbors.

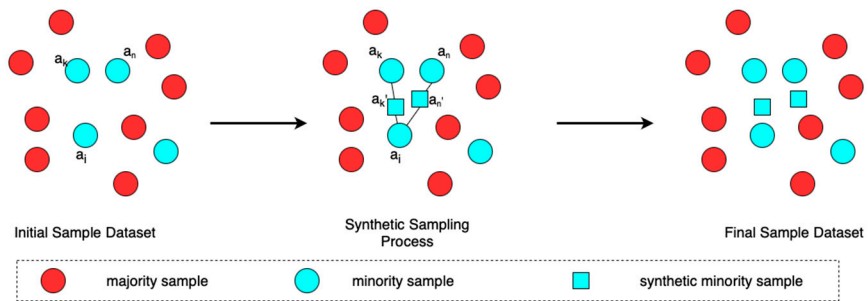

**Figure 2.** Schematic of data augmentation process using SMOTE.

The result of SMOTE is shown in Figure 3.

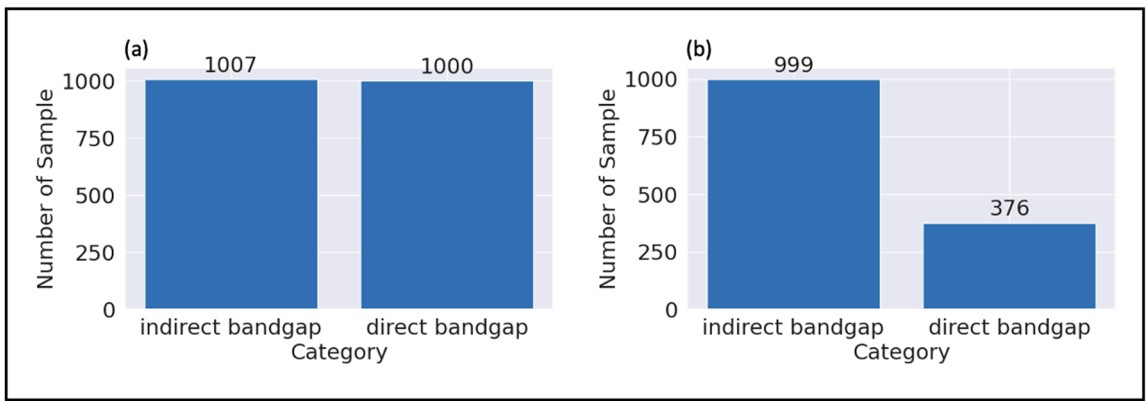

**Figure 3.** The distribution of the nature of bandgap before and after applying SMOTE. (**a**) the distribution of data after applying SMOTE, (**b**) the distribution of data before applying SMOTE.

### 2.2.2. Feature Engineering

The progression of feature engineering is an important phase before applying machine learning algorithms. Feature engineering provides functions to transform and select input variables to enhance the predictive performance of models. One key factor of feature engineering involves dimensionality reduction, which addresses the challenge of high-dimensional datasets. By reducing the number of features while retaining meaningful information, models become more reliable and less complex in terms of dimension. In this endeavor, the Pearson Correlation coefficient emerges as a valuable tool. It enables the quantification of the linear relationship between pairs of features, which is shown in Figure 4. Through the application of Pearson Correlation, features exhibiting strong interdependence are identified, allowing for informed decisions about which features to retain, discard, or transform. Features that display low correlation with the target variable can be pruned, leading to a refined feature set that not only improves computational efficiency but also enriches the predictive capabilities of machine learning models.

### 2.3. Machine Learning Algorithms

In the pursuit of predicting the nature of the bandgap in materials, a fundamental aspect of the methodology involved the application of diverse machine learning algorithms. These algorithms were systematically employed to understand the complex relationships between material properties and bandgap characteristics. A selection of six potential machine learning models was exploited, with each model capturing specific features within the dataset. The summary of the six models is shown in Table 2.

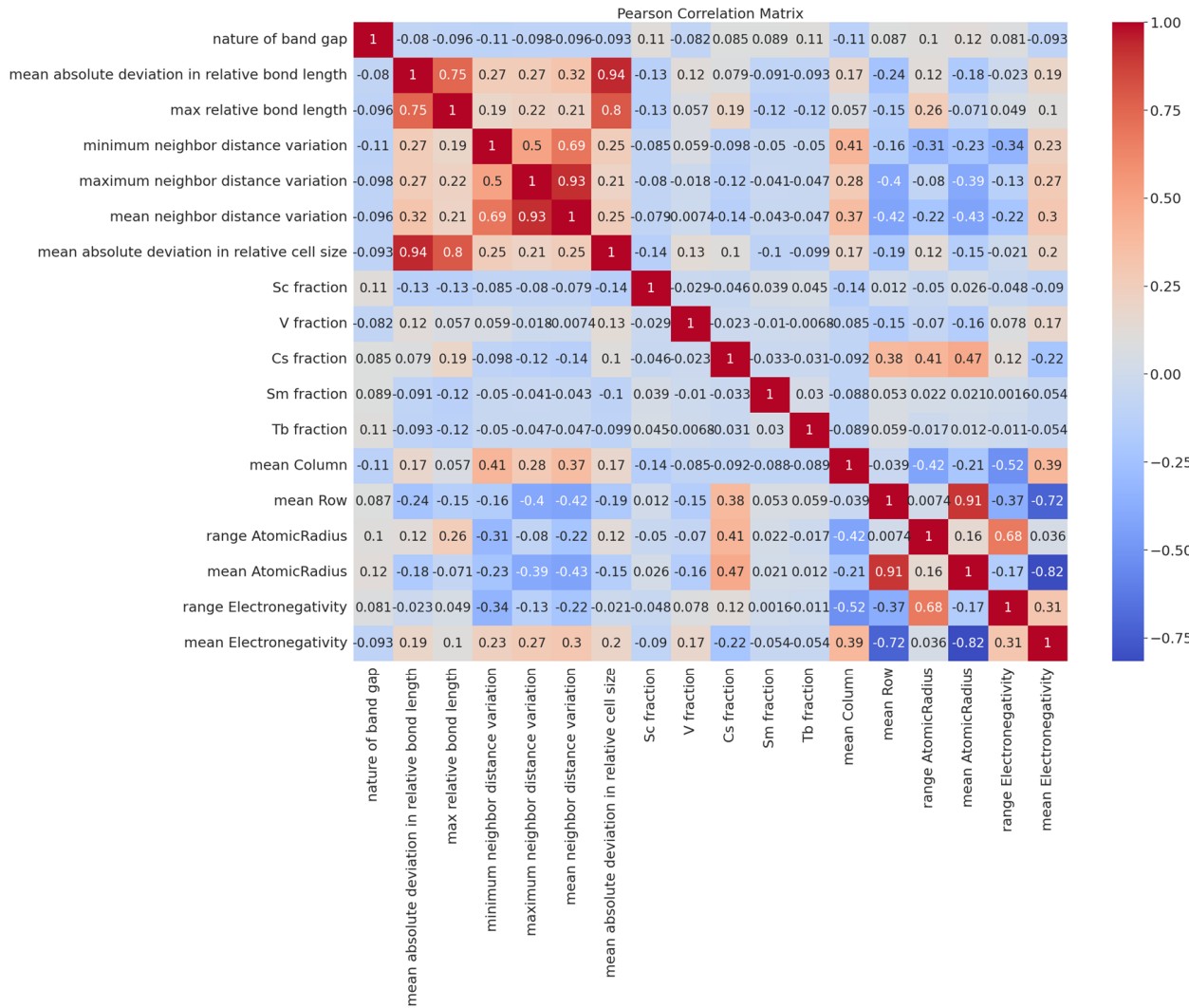

**Figure 4.** The Pearson Correlation results from the 20 most relevant features regarding the nature of the bandgap.

**Table 2.** The summary of mythologies in all machine learning used in the research.

| ML Algorithms | Brief Description | Formular |
|---|---|---|
| Linear Regression (LR) [31] | LR is a fundamental machine-learning technique used for classifying data points into distinct categories. This approach seeks to draw a linear decision boundary that effectively separates different classes in the feature space. | $f(x) = \mathcal{W}^T x + b$<br>where:<br>$x$: input features of data points.<br>$\mathcal{W}$: weight of the vectors.<br>$b$: the bias of the function. |
| Decision Tree (DT) [32] | DT is a versatile and intuitive machine-learning algorithm used for both classification and regression tasks. It resembles a flowchart-like structure, where each internal node represents a decision based on a specific feature, and each leaf node represents a predicted outcome. The algorithm works by recursively partitioning the feature space into subsets based on the values of different attributes. At each step, the attribute that best separates the data is chosen, creating a branching structure. | **Entropy:**<br>$E(S) = -\sum_{\iota=1}^{c} \rho_i \log_2 \rho_i$<br>**Information gain:**<br>$Gain(T, X) = Entropy(T) - Entropy(T, X)$<br><br>**Gini index:**<br>$Gini = 1 - \sum_{\iota=1}^{c} (\rho_i)^2$<br>where:<br>S: the dataset calculated using entropy.<br>ι: the classes in the set, S.<br>ρ: the proportion of data points that belong to class I to the number of total data points in set, S. |

**Table 2.** *Cont.*

| ML Algorithms | Brief Description | Formular |
|---|---|---|
| Random Forest (RF) [33] | RF is a powerful ensemble learning algorithm that leverages the collective wisdom of multiple DTs to enhance prediction accuracy and control overfitting. Each DT in the ensemble is built on a different subset of the data, and its predictions are combined to produce a more robust final prediction. The algorithm introduces randomness at two levels: during data sampling and feature selection. | $\mathcal{F}_{\nabla\{}^{\mathcal{D}}(\S) = \frac{1}{D}\sum\limits_{d=1}^{D} T_d(\S)$ <br> where: <br> $D$: the total number of decision trees in Random Forest <br> $T_d$: the class prediction of the dth Random Forest tree. |
| Support Vector Machine (SVM) [34] | SVM is a versatile machine learning algorithm primarily used for classification tasks, although it can be extended to regression as well. SVM aims to find an optimal hyperplane in a high-dimensional space that best separates data points of different classes. This hyperplane maximizes the margin between the two classes, thereby enhancing the algorithm's generalization capability for new, unseen data. | **Linear SVM (Hard Margin):** <br> $\mathcal{W}^{\mathrm{T}}x + b = 0$ <br> where: <br> $\mathcal{W}$: the weight vector perpendicular to the hyperplane <br> $x$: the feature vector of the data point. <br> $b$: the bias term. <br><br> **Linear SVM (Soft Margin):** <br> $\mathcal{W}^{\mathrm{T}}x + b \geq 1 - \xi_i$ *for the positive class* <br> $\mathcal{W}^{\mathrm{T}}x + b \leq -1 + \xi_i$ *for the negative class* <br> where: <br> $\xi_i$: the slack variable associated with the $i - $ th data point <br><br> Non-Linear SVM (Kernel SVM): <br> $\sum\limits_{i=1}^{N} \alpha_i \gamma_i K_{(\chi_i, \chi)} + b = 0$ <br> where: <br> $N$: the number of support vectors. <br> $\alpha_i$: Lagrange multipliers assciated with support vectors. <br> $\gamma_i$: the class labels of support vectors. <br> $K$: the kernel function that computes the similarity between data points $\chi_i$ and $\chi$ |
| Extreme Gradient Boosting (XGBoost) [35] | XGBoost is an advanced and highly optimized machine learning algorithm used for both classification and regression tasks. XGBoost is an enhanced version of gradient boosting that incorporates regularization techniques to improve predictive accuracy while mitigating overfitting. XGBoost employs an ensemble of decision trees, where each new tree is built to correct the errors of the previous ones. | **Objective Function:** <br> $\mathrm{Obj}(\theta)\mathrm{L} = (\hat{\gamma}_\iota, \gamma_\iota) + \sum\limits_{k=1}^{K} \Omega(f_k)$ <br> where: <br> $\mathrm{Obj}(\theta)$: The objective function to minimize <br> L: the loss term that measures the difference between actual target $\gamma_\iota$ and predict target $\hat{\gamma}_\iota$ <br> ($\Omega$: represent the regularization term, where f_k is the predict of the $k - $ th tree <br> **Individual Tree Prediction:** <br> $f_k(\chi_\iota) = W_p(\chi_\iota)$ <br> where: <br> $f_k(\chi_\iota)$: is the prediction of the k–th tree for the data point $\chi_\iota$ <br> $W_p(\chi_\iota)$: is the weight assigned to the leaf node that data point $\chi_\iota$ <br> **Final Prediction:** <br> $\hat{\mathcal{y}}_i = \sum\limits_{k=1}^{K} f_k(x_i) = \sum\limits_{k=1}^{K} w_q(x_i)$ <br> where: <br> $\hat{\mathcal{y}}_i$: is the final predicted value for the data point $x_i$ <br> $K$: total number of values in each tree |
| Multi-layer Perceptron (MLP) [36] | MLP is a foundational type of artificial neural network (ANN) that excels at capturing complex patterns in data. It consists of multiple layers of interconnected nodes (neurons) organized into an input layer, one or more hidden layers, and an output layer. Each neuron in a layer is connected to every neuron in the subsequent layer. MLP leverages activation functions to introduce non-linearity into its computations, enabling it to model intricate relationships in data. | Neuron Activation: <br> $a_j^{(l)} = f\left(\sum\limits_{i=1}^{n_{l-1}} w_{ij}^{(l)} a_i^{(l-1)} + b_j^{(l)}\right)$ <br> where: <br> $a_j^{(l)}$: the activation of neuron $j$ in layer 1. <br> $w_{ij}^{(l)}$: the weight connecting neuron I in the layer $l - 1$ to neuron $j$ in layer 1. <br> $b_j^{(l)}$: the bias term of neuron $j$ in layer 1 <br> f: the activation function applied to the weighted sum. <br><br> Activation Function (Sigmoid): <br> $f(x) = \frac{1}{1 + e^{-x}}$ |

## 3. Results

The Sections 3 and 3.2 presents the outcomes of the implemented methodologies, shedding light on the effectiveness of the applied techniques in addressing the research objectives. This section highlights quantitative evaluations, such as accuracy, precision, and recall values, that evaluate the performance of the predictive models. Additionally, it delves into qualitative insights, interpreting the significance of specific features and their impact on predicting the nature of the bandgap in inorganic perovskite solar cells.

Before feeding data into machine learning algorithms, the data are split into training and testing sets which are shown in Table 3. After that, in order to choose the most suitable algorithm for predicting the nature of the bandgap, six of the most common models have been selected for evaluation based on accuracy and precision criteria. For measuring the correct quality of the model, all default hyperparameters are kept, ensuring the evaluation is correct. Through the provided information from Table 4, it is easy to see that Random Forest outperformed the others with approximately 82% accuracy. Furthermore, in this study, the confusion matrix [37] is built to evaluate the performance of a classification model. It summarizes the model's predictions by showing the true positive, true negative, false positive, and false negative counts, enabling a detailed assessment of its accuracy and error rates. The detailed result of the confusion matrix is represented in Figure 5. If the prediction labels are correct in Random Forest, it is 95 for the indirect bandgap and 101 for the direct bandgap which is almost 20% higher compared to Linear Regression with just 66 and 79, respectively. In addition, the decision region of the two highest feature correlations, Sc Fraction and mean Atomic Radius, can be used to define the planes of each class.

**Table 3.** The optimization of the hyperparameters used in Random Forest.

| Algorithm | Optimized Hyperparameter |
|---|---|
| Random Forest (RF) | n_estimators: 277, min_samples_split: 5, min_samples_leaf: 1, max_features: 'sqrt', max_depth: 28 |

**Table 4.** The classification report on the performance of the Random Forest algorithm.

|  | Precision | Recall | F1-Score |
|---|---|---|---|
| 0 | 0.85 | 0.85 | 0.85 |
| 1 | 0.86 | 0.86 | 0.86 |
| Accuracy |  |  | 0.86 |
| Macro avg | 0.86 | 0.86 | 0.86 |
| Weighted avg | 0.86 | 0.86 | 0.86 |

In addition to the representation of the confusion matrix, Figure 6 offers a visual depiction of the decision regions in a two-dimensional space for the six distinct machine-learning algorithms. This visualization demonstrates the efficacy of these algorithms in relation to the crucial mean Atomic Radius and Sc fraction, which results in the feature engineering procedure. The outcome describes the domains of data points in which each point is assigned a specific color corresponding to their respective decision regions. This observation reveals that the Decision Tree, Random Forest, and Extreme Gradient Boosting algorithms try to cover all decision regions through relatively fine and contrasting partitions compared to the other algorithms.

### 3.1. Model Evaluation

After validating the data through the six different models, and based on the performance result, the RF algorithm has been chosen for process optimization and is used for the final experiments. To figure out the hyperparameter, the K-fold algorithm with 10 different folds and the Grid Search algorithm, which is a systematic method used for hyperparameter tuning in machine learning and optimization tasks, were used. In the

Grid Search, a predefined set of hyperparameters is specified along with their possible values. The algorithm then exhaustively searches through all possible combinations of these hyperparameter values, evaluating the model's performance using a chosen evaluation metric (e.g., accuracy, precision, F1-score) for each combination. The combination of hyperparameter values that yields the best performance on the validation data is selected as the optimal set of hyperparameters for the model. The result of this algorithm is shown in Table 3.

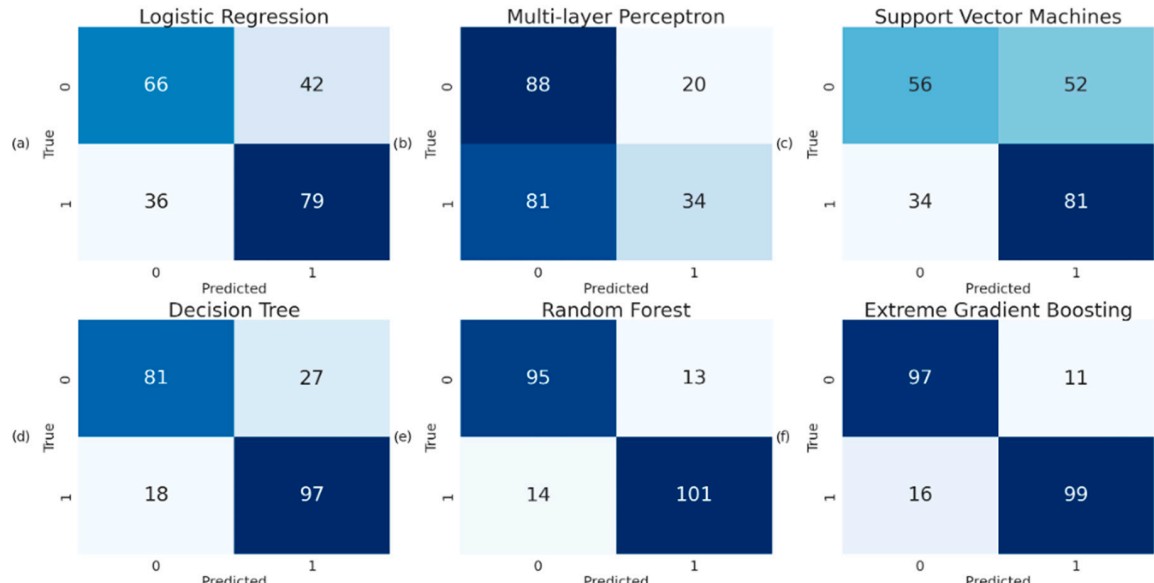

**Figure 5.** The confusion matrix results for the six different machine learning algorithms: (**a**) Logistic Regression, (**b**) Multi-layer Perceptron, (**c**) Support Vector Machine, (**d**) Decision Tree, (**e**) Random Forest, (**f**) Extreme Gradient Boosting.

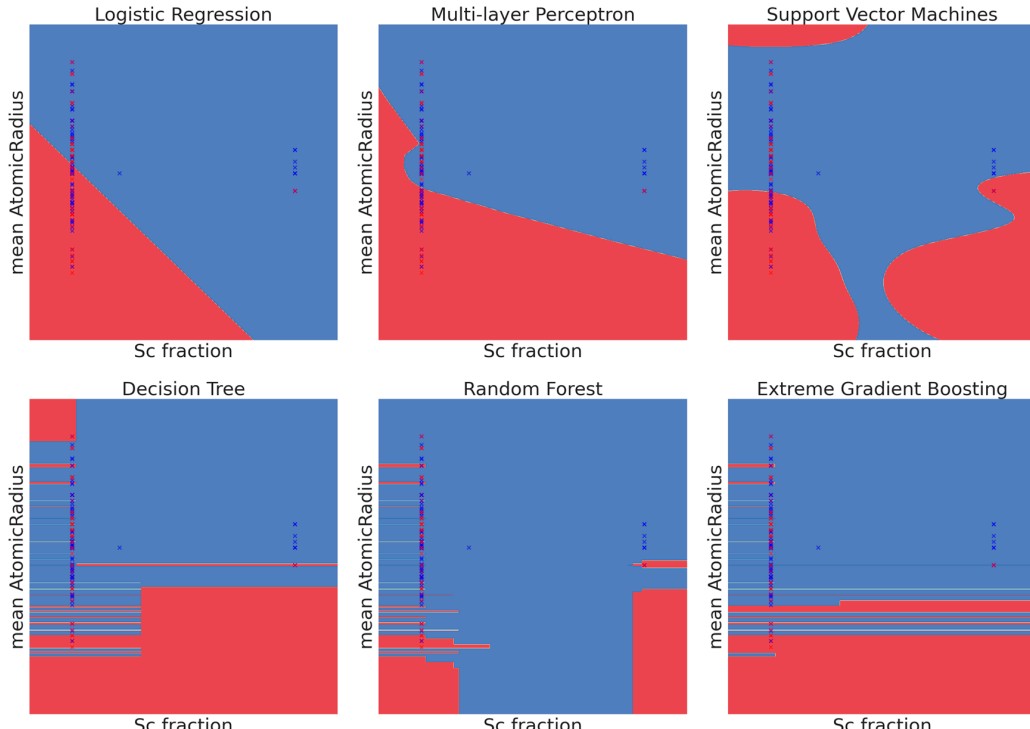

**Figure 6.** The plot of decision region on six machine learning algorithms: red color represents indirect bandgap region and value, and blue color represents direct bandgap region and value.

After hyperparameters are selected, the final experiment with RF was conducted and the result is shown in Table 4.

Table 4 shows the precision, recall, and F1-score results on classification between indirect bandgap (labeled as 0) and direct bandgap (labeled as 1). The table also calculates the average result of each criterion up to 86%. Besides the summary performance table, the ROC curve, Precision-Recall Curve, and Confusion Matrix are also provided to evaluate the model, which is shown in Figure 7.

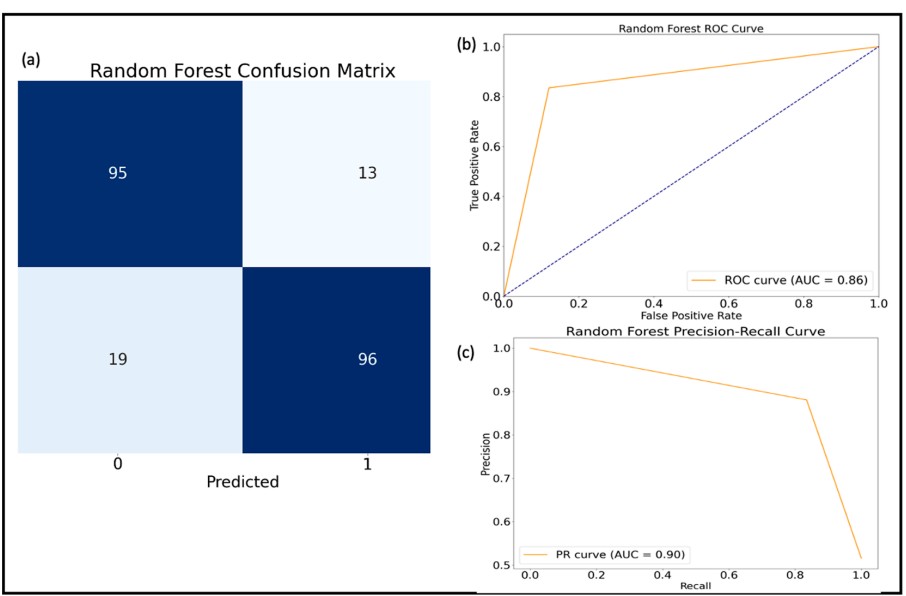

**Figure 7.** The visualization of final experiment on Random Forest. (**a**) Confusion matrix. (**b**) Random Forest ROC Curve. (**c**) Random Forest Precision-Recall Curve.

The higher performance provided by RF algorithms in this experiment demonstrates the potential of using ML algorithms in predicting other properties of ABX$_3$ perovskite in different fields. In addition, it is interesting to show the effectiveness of RF when compared with other algorithms that have been used in the past to predict the properties of ABX$_3$. Gladkikh et al. used kernel ridge regression (KRR) and extremely randomized trees (ERT) to predict the bandgap of perovskites that have a non-zero bandgap to figure out the non-linear relationship between the bandgap and the properties of materials [38]. Ericsson et al. used the Support Vector Regression (SVR) model to find the best prediction of the formation energy with an error rate of Mean Square Error (MAE) at 0.055 eV/atom, and 0.096 eV/atom Root Mean Square Error (RMSE) [39]. Rath [27] et al. have applied Extreme Gradient Boosting (XGBoosting) on the same dataset and achieved an accuracy of 81%.

### 3.2. Discussion

In the conclusive section of the results, a comprehensive exploration is provided through the SHAP (Shapley Additive Explanations) summary plot, providing a deep understanding of the different features that influence the determination of the bandgap nature. As illustrated in Figure 8, the graphical representation describes the relative importance of the top 20 selected input features derived from the Pearson Correlation algorithm. Within this description, the highest impact factor can be seen at the top of the graph, which is minimum neighbor distance variation, and the lowest impact factor can be found at the bottom of the graph which is known as the Tb fraction. Besides the position of each feature being shown, the two different colors, light blue and light pink, also point out the effect of each feature with indirect or direct bandgap. The longer bar in each class also shows the importance of that feature, taking the most change in the decision of the model to classify the nature of the bandgap. Evident from the SHAP summary plot are key insights that illuminate the following discourse:

- The minimum neighbor distance variation shows more importance compared to other features, while the Pearson Correlation shows that the most important feature is the mean Atomic Radius. This observation shows that this metric holds significance as it contributes insights into the structural integrity and electronic properties of the perovskite, which are pivotal factors in determining its properties.
- The presence of the Cs, V, Tb, Sc, and Sm factors [40] does not show the importance to the final prediction of the model, which proves that the amount or concentration of ions in the "A" position of perovskite structure does not make a difference to the result of the experiment which is totally opposite to the initial opinion from Pearson Correlation.
- The role of electronic charge density [41] is lower compared to another feature which is known as the pivotal role in understanding various material properties such as the structure of bandgap, mobility, conductivity, and thermal properties.

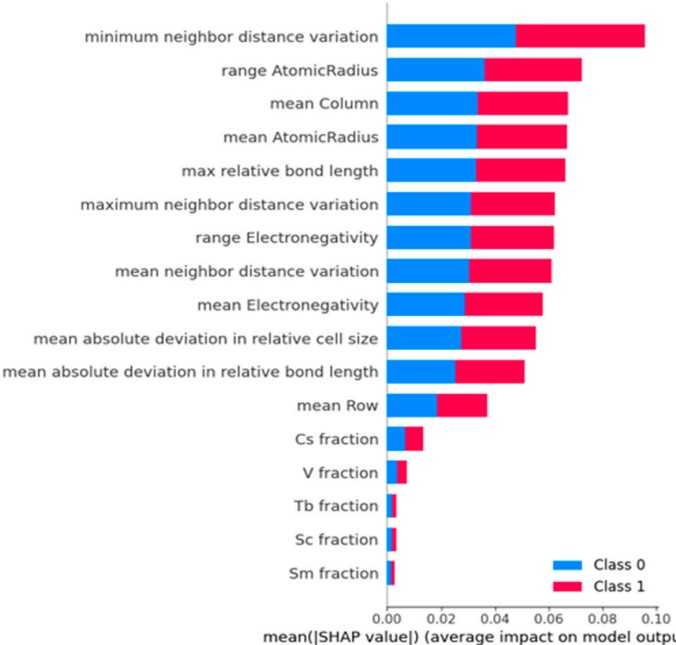

**Figure 8.** A summary plot of SHAP with input features compared to the result of the Random Forest algorithm.

The results from this experiment and the observations above can change the evaluation point of view for future research so that the researcher can have a different approach to selecting the material to maximize the change and reduce experiment time to obtain the direct bandgap.

## 4. Conclusions

Experimental findings highlight the efficacy of the RF algorithm, which achieved an impressive precision rate of 86% on the test dataset. This outcome demonstrates the model's capability to make accurate predictions, rendering it a promising ML model for predicting the nature of the bandgap in inorganic perovskite materials for solar cells. The pivotal role of data augmentation through SMOTE is evidenced by the fact that it effectively mitigated the challenges stemming from class imbalance and data scarcity. By addressing these issues, SMOTE contributes to enhancing the model's performance and robustness. Further supporting this, the SHAP analytic affords a deeper understanding of the intricate relationship between the minimum neighbor distance variation and the nature of the perovskite bandgap. In this regard, the SHAP analysis sheds light on hidden connections and aids in discovering the influential factors affecting the prediction process. These achievements can take advantage of reducing the dependence on traditional methods

and the number of resources needed, have a deeper understanding of the complexities of inorganic perovskite materials for solar cells, and hold promise for future applications in renewable energy technologies.

**Author Contributions:** Conceptualization, T.-C.-H.N. and Y.-U.K.; methodology, T.-C.-H.N. and Y.-U.K.; software, T.-C.-H.N. and Y.-U.K.; validation, T.-C.-H.N., Y.-U.K. and M.S.A.; formal analysis, T.-C.-H.N. and I.J.; investigation, T.-C.-H.N. and I.J.; resources, T.-C.-H.N.; data curation, T.-C.-H.N., Y.-U.K. and M.S.A.; writing—original draft preparation, T.-C.-H.N.; writing—review and editing, M.S.A. and O.-B.Y.; supervision, M.S.A. All authors have read and agreed to the published version of the manuscript.

**Funding:** This research received no external funding.

**Institutional Review Board Statement:** Not applicable.

**Informed Consent Statement:** Not applicable.

**Data Availability Statement:** The data that support the findings of this study are openly available in ML-2021 at https://github.com/smarakrath/MI-2021, (accessed on 27 August 2023), reference number [27].

**Acknowledgments:** This work was supported by the Human Resources Program in Energy Technology of the Korea Institute of Energy Technology Evaluation and Planning (KETEP), granted financial resource from the Ministry of Trade, Industry, and Energy, Republic of Korea (No. 20204010600470). This work was also supported by the Energy International Joint Research Project of the Korea Institute of Energy Technology Evaluation and Planning (KETEP) granted financial from the Ministry of Trade, Industry & Energy, Republic of Korea (No. 20208530050060).

**Conflicts of Interest:** The authors declare no conflict of interest.

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
