# Peer review of "Exploring Data Augmentation and Dimension Reduction Opportunities for Predicting the Bandgap of Inorganic Perovskite through Anion Site Optimization"

_photonics, doi:10.3390/photonics10111232_

Round 1

Reviewer 1 Report

Comments and Suggestions for Authors

This paper presents a machine-learning approach for accurately predicting new inorganic perovskite materials' direct and indirect band gaps. The study focuses on predicting high-performing perovskite materials for solar cells. The paper is well-written, and the ML methods employed are well-documented. This work meets the requirements of articles to be accepted for publication in Photonics. However, prior to publication, a few issues need more clarification.

1. In the abstract, the authors described that the proposed model could predict the band gap of new materials with accuracy and speed. Additionally, it can identify the crucial elements that contribute to this property. However, quantitative values should be added to prove the machine learning model's accuracy.

2. The authors have highlighted the importance of using ML models for perovskite materials in the introduction section. However, the text lacks proper citation of literature on ML-driven optimization of perovskite materials. It is advised that the authors include more relevant literature on perovskite material optimization through ML approaches.

3. In section 2.1, it was explained that “Given the presence of a dataset featuring a multitude of high-dimensional features, the systematic exploration of the dataset becomes an essential process to discover data patterns, duplicate information, and missing values in the dataset.” Authors are encouraged to provide more clarification on duplicate information and missing values in the dataset.  

4. In this work, the skewed distribution poses a significant problem, as the underrepresentation of the "nature of the band gap" class could result in biased model training and hinder accurate predictions for classification tasks. To address this issue, the Synthetic Minority Over-sampling Technique (SMOTE) [25] emerges as a compelling solution. It is better to illustrate the schematic of SMOTE for readers.

5. Figure 4 illustrates the results of the confusion matrix for six different machine learning algorithms. The authors should provide a more detailed explanation of the confusion matrix using relevant literature.

6. It is necessary to improve the clarity of Figure 1.

Comments on the Quality of English Language

Authors should thoroughly review their manuscript for grammatical and typographical errors, as a few sentences are poorly structured.

Author Response

Responses to Reviewer#1

Dated: October 16, 2023

Dear Editor

Thank you for forwarding the reviewer’s comments about our manuscript entitled “Exploring Data Augmentation and Dimension Reduction Opportunities for Prediction the Bandgap of Inorganic Perovskite through Anion Site Optimization” by T.-C.-H. Nguyen et al. You have kindly given us an opportunity for the revision after doing the needful modifications as per reviewer’s suggestions. Please find attached herewith the revised manuscript along with the reply sheet to honorable referees. Now, we have carefully revised and compensated all the reviewer’s comments in the revised manuscript. The corrected parts are highlighted by yellow background in the revised manuscript.

We hope that we have satisfactorily addressed all the valuable comments of the reviewers.

While thanking you in anticipation I am looking forward a favorable reply.

Best Regards

Prof. M. Shaheer Akhtar

Responses to Reviewer

This paper presents a machine-learning approach for accurately predicting new inorganic perovskite materials' direct and indirect band gaps. The study focuses on predicting high-performing perovskite materials for solar cells. The paper is well-written, and the ML methods employed are well-documented. This work meets the requirements of articles to be accepted for publication in Photonics. However, prior to publication, a few issues need more clarification.

1. In the abstract, the authors described that the proposed model could predict the band gap of new materials with accuracy and speed. Additionally, it can identify the crucial elements that contribute to this property. However, quantitative values should be added to prove the machine learning model's accuracy.

Reply: Thanks for valuable suggestions. The quantitative values are now included in the abstract, as “RF yielded the best experimental outcomes according to the following metrics: F1-score, Recall, and Precision, attaining scores of 86%, 85%, and 86%, respectively.”

  1. The authors have highlighted the importance of using ML models for perovskite materials in the introduction section. However, the text lacks proper citation of literature on ML-driven optimization of perovskite materials. It is advised that the authors include more relevant literature on perovskite material optimization through ML approaches.

Reply: Thank you for your necessary comment. Some related works based on ML models for perovskite materials in the introduction section. It is stated as; “X. Cia et al. [37] reported on the optimization of materials discovery by applying ML algorithms to unveil the connection between critical parameters and photovoltaic performance with high-profile MASnxPb1-xI3 perovskite materials. Another work by Q.Tao et al [38] described different ML algorithms to identify different properties of inorganic, hybrid organic-inorganic, and double perovskites to optimize the experiment process during the property’s discovery of new materials. Work reported by J. Li et al [39] employed a dataset comprising 760 perovskites to determine the phonon cutoff frequency and applied six distinct ML algorithms to predict this instrumental variable using features available within the provided database.”

  1. In section 2.1, it was explained that “Given the presence of a dataset featuring a multitude of high-dimensional features, the systematic exploration of the dataset becomes an essential process to discover data patterns, duplicate information, and missing values in the dataset.” Authors are encouraged to provide more clarification on duplicate information and missing values in the dataset.  

Reply: Thanks for valuable suggestion. In addition to the sample shown in Table 1, the entire dataset in this study has been validated to have no missing or duplicate entries during the data preprocessing phase.

  1. In this work, the skewed distribution poses a significant problem, as the underrepresentation of the "nature of the band gap" class could result in biased model training and hinder accurate predictions for classification tasks. To address this issue, the Synthetic Minority Over-sampling Technique (SMOTE) [25] emerges as a compelling solution. It is better to illustrate the schematic of SMOTE for readers.

Reply: Thank you for your review to improve the quality of the paper, to answer your question Figure 2 has been added: to visualize the schematic of SMOTE, and the following content is added to explain the working process of this techniques

The SMOTE works by generating synthetic examples of the minority class by interpolating between existing minority class instances. SMOTE selects a minority class instance, identifies its k-nearest neighbors, and creates synthetic examples by blending the features of the selected instance with those of its neighbors which is visualized in Figure 2.

  1. Figure 4 illustrates the results of the confusion matrix for six different machine learning algorithms. The authors should provide a more detailed explanation of the confusion matrix using relevant literature.

Reply: Thank you for needful suggestion. As per reviewer’s suggestion, we have elaborated the explanation of confusion matrix and cited the reference. It is stated as; “Furthermore, in this study, the confusion matrix [40] is built to evaluate the performance of a classification model. It summarizes the model's predictions by showing the true positive, true negative, false positive, and false negative counts, enabling a detailed assessment of its accuracy and error rates. The detailed result of the confusion matrix is represented in Figure 5, the prediction labels are correct in Random Forest is 95 on indirect bandgap and 101 on direct bandgap which is higher than almost 20% compared to Linear Regression with just 66 and 79, respectively.”

  1. It is necessary to improve the clarity of Figure 1.

Reply: Thank you for needful constructive suggestion. Now the quality of Figure 1 is improved in the revised manuscript.

Reviewer 2 Report

Comments and Suggestions for Authors

In the manuscript titled “Exploring Data Augmentation and Dimension Reduction Opportunities for Prediction the Bandgap of Inorganic Perovskite through Anion Site Optimization”, the authors present a systematical investigation of different machine learning strategies on the prediction of bandgaps of inorganic perovskites. This work is interesting. Authors presented comprehensive discussion to support their conclusions. However, several main issues should be explained.

1.     What is the main idea of this paper. It seems that the authors just compared different machine learning algorithms on the prediction of bandgaps, then, what is the optimal strategy? Maybe the best-performing machine learning strategy for inorganic perovskite is not suitable for other materials.

2.     How to detect the real bandgap of new perovskite materials?

3.     The biggest issue of this work is the poor images with no or small size of the labels, it is not easy to distinguish the difference of each figure. Please consider to revise.

4.     Moreover, the “bandgap” and “band gap” is not consistent in the main text, please make them identical.

5.     Please highlight the importance of this paper on current scenario.

6.     The highest certified efficiencies of perovskite solar cells should be updated.

7.     Many typos, for example ABX3 and PbX6, the numbers should be subscript labeled.

Comments on the Quality of English Language

Okay

Author Response

Responses to Reviewer#2

Dated: October 16, 2023

Dear Editor

Thank you for forwarding the reviewer’s comments about our manuscript entitled “Exploring Data Augmentation and Dimension Reduction Opportunities for Prediction the Bandgap of Inorganic Perovskite through Anion Site Optimization” by T.-C.-H. Nguyen et al. You have kindly given us an opportunity for the revision after doing the needful modifications as per reviewer’s suggestions. Please find attached herewith the revised manuscript along with the reply sheet to honorable referees. Now, we have carefully revised and compensated all the reviewer’s comments in the revised manuscript. The corrected parts are highlighted by yellow background in the revised manuscript.

We hope that we have satisfactorily addressed all the valuable comments of the reviewers.

While thanking you in anticipation I am looking forward a favorable reply.

Best Regards

Prof. M. Shaheer Akhtar

Responses to Reviewer

In the manuscript titled “Exploring Data Augmentation and Dimension Reduction Opportunities for Prediction the Bandgap of Inorganic Perovskite through Anion Site Optimization”, the authors present a systematical investigation of different machine learning strategies on the prediction of bandgaps of inorganic perovskites. This work is interesting. Authors presented comprehensive discussion to support their conclusions. However, several main issues should be explained.

  1. What is the main idea of this paper. It seems that the authors just compared different machine learning algorithms on the prediction of bandgaps, then, what is the optimal strategy? Maybe the best-performing machine learning strategy for inorganic perovskite is not suitable for other materials.

Reply: Thank you for your valuable comment. As we mentioned in this study, the main idea of the work is summarized the application of data augmentation and machine learning algorithms to generate a more diverse dataset from the original inorganic perovskite database and process the classification task to define the nature of the bandgap. Based on results, our model can be recognized as a screening tool for identifying potential materials more quickly and with fewer resource requirements compared to traditional experiments. Furthermore, the final experiment from this paper also provides the hidden relationship between minimum neighbor distance variations to the nature of bandgap are stronger than other properties.

  1. How to detect the real bandgap of new perovskite materials?

Reply: Thanks for the valuable comment. This work mainly described the prediction of the bandgap of inorganic perovskite through anion site optimization by exploring data augmentation and dimension reduction opportunities. The AI prediction was established using the 1528 distinct chemical compounds with the ABX3 formula considering different features. In this work, the Shapley Additive exPlanations (SHAP) was used to define the importance of the input properties towards the nature of bandgap for new material compositions, as presented in Figure 8. It was found that 17 features are the most relevant inputs which are provided through experimental properties followed by performing the trained ML model to depict the direct and incorrect bandgap of new perovskite materials. It hopes that the findings of our designed ML model might be provided the importance information for achieving the desirable bandgap of new perovskite materials.

  1. The biggest issue of this work is the poor images with no or small size of the labels, it is not easy to distinguish the difference of each figure. Please consider to revise.

Reply: Thank you for needful constructive comments. Qualities of figures have been updated with the text size for good readability in the revised manuscript.

  1. Moreover, the “bandgap” and “band gap” is not consistent in the main text, please make them identical.

Reply: Thank you for needful constructive comments. The synchronization of “bandgap” has been done in the revised manuscript.

  1. Please highlight the importance of this paper on current scenario.

Reply: Thanks for your constructive suggestion. In the introduction, the importance of this work is emphasized in the revised manuscript.

  1. The highest certified efficiencies of perovskite solar cells should be updated.

Reply: Thanks for your constructive suggestion. The highest certified efficiencies of perovskite solar cells are now included in the revised manuscript.

  1. Many typos, for example ABX3 and PbX6, the numbers should be subscript labeled.

Reply: Thank you for needful suggestion. We have thoroughly checked the manuscript and corrected the typo-errors.

Reviewer 3 Report

Comments and Suggestions for Authors

The manuscript on "Exploring Data Augmentation and Dimension Reduction Op- 2 portunities for Prediction the Bandgap of Inorganic Perovskite through Anion Site Optimization" is well writtem and quiet significant for the organic perovskites community. I have few minor comments 

Figure 3 and 5  is hard to read the axis 

References needs to be updated 

What does the error represent here? Authors are required to explain how the errors bars are obtained. 

Comments on the Quality of English Language

None

Author Response

Responses to Reviewer#3

Dated: October 16, 2023

Dear Editor

Thank you for forwarding the reviewer’s comments about our manuscript entitled “Exploring Data Augmentation and Dimension Reduction Opportunities for Prediction the Bandgap of Inorganic Perovskite through Anion Site Optimization” by T.-C.-H. Nguyen et al. You have kindly given us an opportunity for the revision after doing the needful modifications as per reviewer’s suggestions. Please find attached herewith the revised manuscript along with the reply sheet to honorable referees. Now, we have carefully revised and compensated all the reviewer’s comments in the revised manuscript. The corrected parts are highlighted by yellow background in the revised manuscript.

We hope that we have satisfactorily addressed all the valuable comments of the reviewers.

While thanking you in anticipation I am looking forward a favorable reply.

Best Regards

Prof. M. Shaheer Akhtar

Responses to Reviewer

The manuscript on "Exploring Data Augmentation and Dimension Reduction Opportunities for Prediction the Bandgap of Inorganic Perovskite through Anion Site Optimization" is well written and quite significant for the organic perovskites community. I have few minor comments. 

  1. Figure 3 and 5 is hard to read the axis.

Reply: Thank you for needful constructive comments. The axis of Figure 3 and 5 have been modified and make them more visible.

  1. References needs to be updated.

Reply: Thank you for needful constructive comments. The references have updated in the revised manuscript.  

  1. What does the error represent here? Authors are required to explain how the errors bars are obtained. 

Reply: Thank you for needful constructive comments. In Figure 6, the distribution of data is not the error representation. Actually, it is the point’s distribution of the decision regions for direct and indirect bandgaps via six machine learning algorithms. Now a brief explanation has added in the revised manuscript as; “In addition to the representation of the confusion matrix, Figure 6 offers a visual depiction of decision regions in a two-dimensional space for six distinct machine-learning algorithms. This visualization demonstrates the efficacy of these algorithms in relation to the crucial mean atomic radius and Sc fraction, which results the feature engineering procedure. The outcome describes the domains of data points in which each point is assigned a specific color corresponding to their respective decision regions. This observation reveals that the Decision Tree, Random Forest, and Extreme Gradient Boosting algorithms try to cover all decision regions through relatively fine and contrasting partitions compared to the other algorithms.”

Round 2

Reviewer 2 Report

Comments and Suggestions for Authors

Okay for publish.